# Structure/Function Studies on the Activation Motif of Two Non-Mammalian Mrap1 Orthologs, and Observations on the Phylogeny of Mrap1, Including a Novel Characterization of an Mrap1 from the Chondrostean Fish, *Polyodon spathula*

**DOI:** 10.3390/biom12111681

**Published:** 2022-11-12

**Authors:** Robert M. Dores, Greer McKinley, Audrey Meyers, Morgan Martin, Ciaran A. Shaughnessy

**Affiliations:** Department of Biological Sciences, University of Denver, Denver, CO 80210, USA

**Keywords:** MRAP1, MC2R, evolution, paddlefish

## Abstract

In derived bony vertebrates, activation of the melanocortin-2 receptor (Mc2r) by its ACTH ligand requires chaperoning by the Mc2r accessory protein (Mrap1). The N-terminal domain of the non-mammalian tetrapod MRAP1 from chicken (c; *Gallus gallus*) has the putative activation motif, W^18^D^19^Y^20^I^21^, and the N-terminal domain in the neopterygian ray-finned fish Mrap1 from bowfin (bf; *Amia calva*) has the putative activation motif, Y^18^D^19^Y^20^I^21^. The current study used an alanine-substitution paradigm to test the hypothesis that only the Y^20^ position in the Mrap1 ortholog of these non-mammalian vertebrates is required for activation of the respective Mc2r ortholog. Instead, we found that for cMRAP1, single alanine-substitution resulted in a gradient of inhibition of activation (Y^20^ >> D^19^ = W^18^ > I^21^). For bfMrap1, single alanine-substitution also resulted in a gradient of inhibition of activation (Y^20^ >> D^19^ > I^21^ > Y^18^). This study also included an analysis of Mc2r activation in an older lineage of ray-finned fish, the paddlefish (p), *Polyodon spathula* (subclass Chondronstei). Currently no *mrap1* gene has been found in the paddlefish genome. When *pmc2r* was expressed alone in our CHO cell/cAMP reporter gene assay, no activation was observed following stimulation with ACTH. However, when *pmc2r* was co-expressed with either *cmrap1* or *bfmrap1* robust dose response curves were generated. These results indicate that the formation of an Mc2r/Mrap1 heterodimer emerged early in the radiation of the bony vertebrates.

## 1. Introduction

Studies on the activation of human (h) MC2R, one of the five vertebrate melanocortin receptors (MCRs) (i.e., MC1R, MC2R, MC3R, MC4R, MC5R), indicate that MC2R has the most complex activation mechanism [1,2,3]. The hMC2R cannot be functionally expressed in most non-adrenal mammalian cell lines [4], and unlike the other vertebrate MCRs, hMC2R can only be activated by adrenocorticotropin (ACTH), but not by any of the melanocyte-stimulating hormone ligands (i.e., α-MSH, β-MSH or γ-MSH [1,5]). Similar observations have been made for non-mammalian tetrapod Mc2r orthologs, and neopterygian fish Mc2r orthologs [6,7]. The discovery of the accessory protein, MRAP1 (melanocortin-2 receptor accessory protein 1), which was first described in humans [8], has helped to resolve some of the novel features associate with MC2R activation. A defining feature of the Mc2r of osteichthyan vertebrates (i.e., neopterygian bony fishes, amphibians, reptiles, birds, and mammals) is the obligatory interaction with the accessory protein Mrap1 to facilitate the trafficking and activation of Mc2r orthologs [7].

Mrap1 is a single pass transmembrane protein that serves as a chaperone for most Mc2r orthologs [2,3,6,7]. Initial studies of this accessory protein have used mouse (*Mus musculus*; m) Mrap1 as a model system to understand the dynamics of the interaction with hMC2R [2]. These studies found that mMrap1 is a homodimer with reverse topology [9] that forms a heterodimer with hMC2R [10]. Within the N-terminal domain of mMrap1 there is the amino acid motif, L^18^D^19^Y^20^I^21^, [11] that facilitates activation of hMC2R following an ACTH binding event. Alanine substitution at all four positions in the “activation motif” completely blocks activation of hMC2R, but did not interfere with the trafficking of the receptor from the ER to the plasma membrane [11]. However, single-alanine substitution of the residues in the mMrap1 “activation motif” indicated that only the Y^20^/A^20^ mutation significantly disrupted activation [11]. The latter study called into question the role of the other residues in the mMrap1 “activation motif”, and whether the “activation motifs” of all osteichthyan Mrap1 orthologs behave in a similar manner.

Putative activation motifs are present in the N-terminal domain of the MRAP1 ortholog of the non-mammalian tetrapod, Gallus gallus (i.e., chicken; c), and the neopterygian fish Amia calva (bowfin; bf) [12]. For both species, these motifs fit the criteria for an Mrap1 activation motif (hydrophobic amino acid (δ), aspartic acid (D), tyrosine (Y), hydrophobic amino acid (δ)) [7]. To test the hypothesis that only Y20 in the N-terminal region of Mrap1 is required for activation of all osteichthyan vertebrate Mc2rs, rather than a mammalian-specific phenomenon, we used an alanine substitution paradigm to evaluate the contribution of each amino acid in the putative activation motifs of cMRAP1 and bfMrap1 to facilitate the activation of their respective Mc2r orthologs following stimulation with ACTH.

The second objective of this study was to gain some insight as to when, during the radiation of the osteichthyan vertebrates, Mc2r became dependent on Mrap1 for activation and exclusively selective for ACTH. Previous studies on tetrapods and neopterygian fishes have indicated that the respective Mc2r orthologs of these osteichthyan vertebrates require interaction with Mrap1 for activation, and that these Mc2r orthologs are exclusively selective for activation by ACTH (for review see [7]). However, cartilaginous fish Mc2r orthologs can be activated by either ACTH or α-MSH-related ligands [7]. Additionally, the primary function of Mrap1 in elasmobranchs appears to be to facilitate the trafficking of the elasmobranch Mc2r ortholog to the plasma membrane of interrenal cells [13]. Since the ancestral cartilaginous fishes and the ancestral osteichthyan fishes evolved from a common ancestral gnathostome [14], an analysis of the pharmacological properties of Mc2r from a basal osteichthyan group such as the chondrostean fishes would be informative. The chondrostean fishes include the bichir, ropefish, the paddlefishes, and the various species of sturgeon [15]. The fossil record indicates that the chondrostean fishes are the oldest extant group of ray-finned fishes [16]. For this study we focused on the Mc2r ortholog of the paddlefish (p) (*Polyodon spathula*). Currently no *mrap1* gene has been found in the paddlefish genome. This observation raised the possibility that pMc2r might be an Mrap1-independent Mc2r ortholog. To test that hypothesis and to determine whether pMc2r could be activated by either ACTH or an MSH-sized ligand, *pmc2r* was transiently expressed in Chinese hamster ovary (CHO) cells, and the efficacy of stimulation with melanocortin ligands was evaluated using a cAMP reporter gene assay.

## 2. Materials and Methods

### 2.1. Mc2r and Mrap1 Sequences

Several Mc2r orthologs and Mrap1 orthologs were used in this study. The nucleotide sequences of bowfin (bf) *mc2r* and *mrap1* were obtained from the bowfin genome project [17]. The identification number for *bfmc2r* is (LOC: 24677757-24678686) and the identification number for *bfmrap1* is (AMCT00016091). The nucleotide sequences of chicken (c) *MC2R* (Accession No. XM_015282470.1) and *cMRAP1* (XR_001470382.1) were obtained from GenBank. Several alanine-substituted mutants of *bfmrap1* and *cMRAP1* were made to target the residues of the putative activation motifs (Appendix A), and the cDNA constructs for each *cmrap1* alanine mutant and *bfmrap1* alanine mutant were synthesized by GenScript (Piscataway, NJ, USA). Additionally, the nucleotide sequences for whale shark (ws) *mrap1* (XP_020375601) and *P. spathula* (p) *mc2r* (XM_041240699.1) were obtained from GenBank. Each cDNA sequence was individually inserted into a pcDNA3+ expression vector (GenScript; Piscataway, NJ, USA). The cAMP reporter gene construct CRE-Luciferase [18] was provided by Dr. Patricia Hinkle (University of Rochester, NY, USA).

### 2.2. Melanocortin Peptides

For the cAMP reporter gene assays, transfected cells were either stimulated with human adrenocorticotropic hormone [hACTH(1-24)] or α-MSH (N-acetyl-SYSMEHFRWGKPV-amide) purchased from Sigma-Aldrich Inc. (St. Louis, MO, USA). The melanocortin peptides were used at concentrations from 10^−12^ M to 10^−6^ M.

### 2.3. cAMP Reporter Gene Assay

The cAMP reporter gene assay was done in Chinese hamster ovary (CHO) cells (ATCC, Manassas, VA, USA) grown in Kaighn’s Modification of Ham’s F12K media (ATCC) and supplemented with 10% fetal bovine serum, 100 unit/mL penicillin, 100 µg/mL streptomycin, 100 µg/mL normocin. The CHO cells were maintained in a humidified incubator with 95% air and 5% CO_2_ at 37 °C. This cell line was selected because the CHO cells do not express endogenous *mcr* genes [4,9], or endogenous *mrap* genes [19].

To perform the cAMP reporter gene assay, species-specific *mc2r* cDNA (i.e., *cMC2R*, *bfmc2r*, *pmc2r*) (10 nmol/transfection) were either expressed alone, with a *mrap1* cDNA or with an alanine-mutant *mrap1* cDNA (30 nmol/transfection), and the *cre-luc* cDNA construct (83 nmol/transfection; cAMP CRE-Luciferase construct; 18) in 3.0 × 10^6^ CHO cells as described previously [20]. The transient transfections were done using the Amaxa Cell Line Nucleofector II system (Lonza, Basel, Switzerland), solution T (Lonza), and program U-23. The transfected cells were plated in a white 96-well plate (Costar 3917, Corning In., Kennebunk, ME, USA) at a final density of 1 × 10^5^ cells/well. After 48 h, the transfected cells were stimulated with various concentrations (10^−12^ M to 10^−6^ M) of hACTH(1-24) or α-MSH in serum-free CHO media for 4 h at 37 °C. Following the incubation period, the stimulating media was removed, and 100 µL of luciferase substrate reagent (Bright GLO; Promega, Madison, WI, USA) was aliquoted into each well. After a 5 min incubation period at room temperature, the luminescence from each well was immediately measured using a Bio-Tek Synergy HT plate reader (Winooski, VT, USA). To determine the background levels of cAMP production, transfected CHO cells were stimulated with serum-free CHO media containing no melanocortin peptide for the 4 h incubation period, and the average background luminescence reading for each assay was subtracted from the luminescence readings of ligand-stimulated assays. All assays were performed in triplicate.

### 2.4. Statistical Analysis

The dose response curves for each assay were fitted to the Michaelis-Menton equation to obtain EC_50_ and V_max_ values. Statistical analysis of the EC_50_ and V_max_ values for the dose response curves were performed using either unpaired two-tailed Student’s *t*-test assuming equal variance or one-way ANOVA followed by Tukey’s multi-comparison test using GraphPad Prism 2 software (GraphPad Inc., La Jolla, CA, USA). Significance was set at *p* ≤ 0.05. Unless otherwise noted, all data are presented as mean ± SEM with *n* = 3. Graphs were prepared using Kaleidograph software (www.synergy.com, accessed on 11 February 2021).

## 3. Results

### 3.1. Analysis of the W^18^D^19^Y^20^I^21^ Motif in cMRAP1

A putative activation motif is present in the N-terminal domain of both the cMRAP1 and bfMrap1 orthologs (Figure 1).

Simultaneous alanine substitution at all positions in the W^18^D^19^Y^20^I^21^ motif of cMRAP1 completely blocked the activation of cMC2R following stimulation with hACTH(1-24) (Figure 2A). To evaluate the role of each residue in the W^18^D^19^Y^20^I^21^ motif, single alanine mutants for each residue of the activation motif of cMRAP1 were examined. Co-expression of cMC2R with either cMRAP1-W^18^/A or cMRAP1-D^19^/A resulted in an approximately 10-fold decrease in ligand sensitivity to stimulation by hACTH(1-24) (i.e., increase in EC_50_), although neither was a statistically significant decrease in sensitivity (Table 1). However, a comparison of V_max_ values indicated that co-expression of cMC2R with either cMRAP1-W^18^/A or cMRAP1-D^19^/A resulted in a decrease in maximal cAMP production. The V_max_ for the wildtype cMRAP1 positive control (*cMC2R/cMRAP1* transfection) was 6926 ± 282 activity units, which was significantly higher than the Vmax values for the *cMC2R/cMRAP1-W18/A18* (3715 ± 259 activity units) and *cMC2R/cMRAP1-D^19^/A* (2864 ± 234 activity units) alanine substitution assays (*p* < 0.001; one-way ANOVA). These values represent decreases in Vmax from the positive control of 46% and 60%, respectively.

Next, the cMC2R activation was examined when co-expressed with either the cMRAP1-Y^20^/A or the cMRAP1-I^21^/A mutants. Whereas the positive control of cMC2R co-expressed with wildtype cMRAP1 again generated a robust dose-dependent activation response when stimulated with hACTH(1-24), alanine substitution at Y^20^ in cMRAP1 (*cMC2R/cMRAP1-Y^20^/A^20^* transfected cells) effectively blocked activation of cMC2R, with only marginal stimulation occurring at the highest dose (10^−6^ M) of the hACTH(1-24) ligand (Figure 2B). Co-expression of cMC2R with the cMRAP1-I^21^/A resulted in an increase in EC_50_ of approximately 10-fold compared to the wildtype cMRAP1 positive control, but this increase was not statistically significant (Table 1). Thus. it appears that alanine substitution at the I^21^ position of cMRAP1 has only a minimal effect on cMC2R activation. Collectively, our results indicate that the hierarchy of importance of residue positions in the activation motif of cMRAP1 is Y^21^ >> D^19^ = W^18^ > I^21^.

### 3.2. Analysis of the Y^18^D^19^Y^20^I^21^ Motif in bfMrap1

The structure/function analysis of the Y^18^ D^19^ Y^20^ I^21^ motif of bfMRAP1 is presented in Figure 3. Alanine substitution of all 4 residue positions of the bfMrap1 activation motif completely blocked the activation of bfMc2r (Figure 3A) in a manner identical to what was observed for the 4-residue alanine substitution mutant of cMRAP1 (Figure 2A). Single-alanine substitution at Y^18^ of bfMrap1 had a minimal effect on the sensitivity of bfMc2r to stimulation by hACTH(1-24) (Table 1) and resulted in no difference in V_max_ compared to the wildtype bfMrap1 positive control. Alanine substitution at D^19^ of bfMrap1 also did not have a statistically significant effect on the EC_50_ as compared to the positive control (Table 1). However, a comparison of the V_max_ values for the positive control (60,340 ± 2251 activity units) and the D^19^/A bfMrap1 mutant (31,406 ± 2467 activity units) revealed that co-expression of bfMc2R with the mutant bfMrap1 resulted in a 48% decrease in activation (*p* < 0.001; one-way ANOVA) (Figure 3A).

The most dramatic effect was observed following alanine substitution at Y^20^ of bfMrap1. Sensitivity of bfMc2r to stimulation by hACTH(1-24) when co-expressed with the Y^20^/A bfMrap1 mutant was nearly a 1000-fold lower than when co-expressed with the wildtype bfMrap1 positive control (Figure 3B, Table 1). A comparison of V_max_ indicates that maximal cAMP production in the *bfmc2r/bfmrap1y^20^/a* transfected cells (4605 ± 376 activity units) was nearly 80% lower than the *bfmc2r/bfmrap1* (positive control) transfected cells (22,101 ± 997 activity units) (*p* < 0.001; one-way ANOVA). Alanine substitution at the I^21^ position in bfMrap1 resulted in a significant, 10-fold decrease in sensitivity to stimulation by ACTH(1-24) as compared to the positive control (Figure 3B, Table 1). The V_max_ for the *bfmc2r/bfmrap1-i^21^/a* transfected cells (16,512 ± 689 activity units) was 25% lower than the V_max_ of the positive control *(p* < 0.01: one-way ANOVA). Together, these results indicate that the hierarchy of importance of residue positions in the activation motif of bfMrap1 is Y^20^ >> D^19^ > I^21^ > Y^18^.

### 3.3. Pharmacological Properties of Paddlefish Mc2r

The paddlefish is a member of Class Chondrostei, the most basal lineage of extant Actinopterygii (ray-finned fishes) [15]. The apparent absence of an *mrap1* gene in the paddlefish genome [21], raised the possibility that paddlefish (p) Mc2r may be an Mrap1-independent Mc2r ortholog and may respond to stimulation by either ACTH or MSH-sized ligands in a manner similar to some cartilaginous fish Mc2r orthologs [7]. To determine whether the activation of pMc2r is dependent on interaction with Mrap1, *pmc2r* was expressed alone in CHO cells and stimulated with hACTH(1-24). The positive control for this experiment was to transfect CHO cells with *bfmc2r* and *bfmrap1*. In the first experiments, the *bfmc2r/bfmrap1* transfected cells (positive control) demonstrated an expected activation response when stimulated with hACTH(1-24), but the *pmc2r* transfected cells did not exhibit any apparent receptor activation (Figure 4A). In the next experiment, *pmc2r* was separately co-expressed with two osteichthyan Mrap1 orthologs (*cMRAP1* or *bfmrap1*), or a cartilaginous fish Mrap1 ortholog (whale shark; ws; *wsmrap1*). The *pmc2r/wsmrap1* transfected cells did not respond to stimulation by hACTH(1-24) (Figure 4B). However, both the *pmc2r/bfmrap1* transfected cells and the *pmc2r/cMRAP1* transfected cells responded in nearly an identical manner when stimulated with hACTH(1-24) (Figure 4B). The EC_50_ values for pMc2r co-expressed with either cMRAP1 or bfMrap1 were 3.1 × 10^−10^ ± 1.1 × 10^−10^ M and 2.9 × 10^−10^ ± 1.3 × 10^10^ M, respectively. To examine the importance of the activation motif in bfMrap1 for the activation of pMc2r, CHO cells were either co-transfected with *pmc2r*/*bfmrap1* (positive control) or *pmc2r*/*bfmrap1-y^20^/a^20^*. In this experiment, pMc2r was unable to be activated when co-expressed with the Y^20^/A^20^ bfMrap1 mutant (Figure 4C). Finally, to examine ligand selectivity of pMc2r *pmc2r/bfmrap1* transfected cells were stimulated with either hACTH(1-24) or N-Acetyl-ACTH(1-13)-amide (α-MSH). Here, pMc2r exhibited only a marginal response to α-MSH at the highest ligand concentration (10^−6^ M), compared to a robust activation by the hACTH(1-24) (Figure 4D).

## 4. Discussion

For osteichthyan vertebrates, the functionality of the hypothalamus/pituitary/adrenal-interrenal (HPA/HPI) axis is dependent on the anterior pituitary hormone ACTH selectively activating the “ACTH” receptor (Mc2r) on steroidogenic cells. The steroidogenic cells will then release glucocorticoids to restore homeostasis after a chronic stress event [22]. As summarized in Figure 5, several studies on tetrapod Mc2r orthologs support the conclusion that the ancestral tetrapods had an Mc2r ortholog that was Mrap1-dependent and exclusively selective for ACTH (Figure 5) and the activation process requires that Mc2r forms a heterodimer with the accessory protein Mrap1 (for review see [7]).

For the tetrapods, the first function of Mrap1 is to facilitate the trafficking of the Mc2r ortholog from the endoplasmic reticulum to the plasma membrane [2,3]. Once the Mc2r/Mrap1 heterodimer is at the plasma membrane, the N-terminal domain of Mrap1 plays a role in activation of Mc2r (discussed in greater detail below). For the neopterygian fishes (i.e., the more recent ray-finned fish lineages), such as the bowfin (present study and [12], gar [23], rainbow trout [24], seabream [25], and zebrafish [26], Mc2r orthologs have the same ligand specificity and Mrap1 dependence as the tetrapods (Figure 5). These observations would suggest that the derived functional traits of Mc2r (i.e., ACTH exclusivity and Mrap1 dependence) were also present in the ancestral Actinopterygii and ancestral Sarcopterygii (Figure 5). Currently, no analyses have been performed on the Mc2r ortholog for the oldest lineages of the Sarcopterygii, the coelacanths and lungfishes, which would add much needed support to this hypothesis. However, in the present study we evaluated the pharmacological properties of an Mc2r ortholog from a basal actinopterygian, the paddlefish. We demonstrated that the pMc2r requires co-expression with an osteichthyan Mrap1 ortholog for activation. Hence, it would be reasonable to conclude that there is an *mrap1*-like gene in the paddlefish genome that has not be annotated as of yet. In addition, pMc2r could be activated by hACTH(1-24) at physiological relevant concentrations, but not by α-MSH (Figure 4D). During the preparation of this manuscript, a study on another chondrostean fish, the bichir *Polypterus senegalus*, observed that the Mc2r ortholog of that species has the same properties as paddlefish Mc2r [12]. Collectively, these observations would suggest that the osteichthyan fish lineage that.

Since the ancestral osteichthyan fishes and the ancestral cartilaginous fishes evolved from a common gnathostome ancestral lineage [14,27], the preceding conclusion raises a number of questions with respect to origins of the pharmacological properties of cartilaginous fish Mc2r orthologs. Currently, three cartilaginous fish Mc2r orthologs have been analyzed: a holocephalon, the elephant shark (*Callorhinchus milii*) [19,28] and two elasmobranchs, the red stingray (*Dasyatis akajei*) [29] and the whale shark (*Rhincodon typus*) [13]. In brief, elephant shark Mc2r is an Mrap1-independent Mc2r ortholog that does not require Mrap1 for trafficking or activation [19,28]. This Mc2r ortholog can be activated by either ACTH or ACTH(1-13)amide (the non-acetylated analog of α-MSH) with near equal efficacy [28]. Whether these properties are unique to the elephant shark or common to other holocephalon-related fishes has not been evaluated. For the two elasmobranch Mc2r orthologs, co-expression with a cartilaginous fish Mrap ortholog (Mrap1 or Mrap2) appears to be necessary to facilitate membrane trafficking [13], yet the requirement (if any) of Mrap chaperoning for Mc2r activation is less apparent and still being evaluated. It is clear that both elasmobranch Mc2r orthologs can be activated by either ACTH or ACTH(1-13)amide at physiologically relevant concentrations of each ligand [13,29]. Assuming that the pharmacological properties of the cartilaginous fish Mc2r orthologs reflect the properties of the Mc2r orthologs of the ancestral gnathostomes, a considerable divergence has occurred in ligand selectivity and interaction with Mrap1 during the early radiation of the ancestral cartilaginous fishes and the ancestral osteichthyan fishes [14] (Figure 5). This functional divergence corresponds to the relatively low primary sequence identity between elephant shark Mc2r and mammalian MC2R orthologs [30]. A recent phylogenic analysis of a broad spectrum of cartilaginous fish and osteichthyan Mc2r orthologs indicated that the number of amino acid substitutions is much higher for the Mc2r orthologs as compared to other Mcr paralogs, such as Mc5r [23]. The rate of substitution for the *mc2r* gene is surprising given the apparent importance of Mc2r function in the HPA/HPI axis of vertebrates [31]. Importantly, alignment of the primary sequences of Mc2r orthologs does not provide enough information to readily reveal why cartilaginous fish Mc2r orthologs can be activated by both ACTH and α-MSH, whereas the osteichthyan Mc2r orthologs are exclusively selective for ACTH. A comparative study on the three-dimensional structure of cartilaginous and osteichthyan Mc2r orthologs could perhaps address this question.

The timing of the bifurcation of vertebrate Mc2r orthologs after the divergence of the ancestral cartilaginous fishes and the ancestral osteichthyan fishes appears to parallel changes that have occurred in the *mrap1* gene, and these changes have had important ramifications for the interactions that occur within the Mc2r/Mrap1 heterodimer. In this representative set of osteichthyan Mrap1 orthologs (Figure 6) there is considerable primary sequence identity/similarity in the transmembrane domain (85%) which is consistent with the role that this domain plays in the trafficking of Mc2r orthologs. In addition, the primary sequence identity/similarity in the N-terminal domain is 62%, highlighted by the δDYδ motif present in each sequence.

As demonstrated in this study, the identity and roles of the flanking hydrophobic amino acids (δ) vary by species, but the overall effect of these residues on Mc2r activation appears to be minimal (Figure 2 and Figure 3). However, the presence of flanking hydrophobic residues is a consistent structural feature of osteichthyan Mrap1 and demarcates the location of an activation motif in the osteichthyan Mrap1 orthologs. For both cMRAP1 and bfMrap1, the integrity of Y^20^ is critical for activation of their respective Mc2r orthologs (Figure 2B and Figure 3B). This observation is consistent with observations made for mouse (m) Mrap1 [11], zebrafish (zf) Mrap1 and rainbow trout (rt) Mrap1 [24]. However, D^19^ in cMRAP1 and bfMrap1 also appears to play a role in the activation process (Figure 2A and Figure 3A), and similar observations were also made for zfMrap1 and rtMrap1 [24]. A recent study on the interaction between hMC2R and mMrap1 shed light on the mechanism(s) by which the N-terminal domain of MRAP1 affects activation of MC2R [32]. It appears that the activation motif in the N-terminal domain makes contact with extracellular loop 2 of MC2R and that this interaction is required for activation of the receptor by ACTH [32]. Classical binding studies are needed to further elucidate the mechanism(s) by which Mrap1 modulates Mc2r in support of this hypothesis.

In contrast to the osteichthyan Mrap1 orthologs, the two cartilaginous Mrap1 sequences lack the δ-D-Y-δ motif in their respective N-terminal domains (Figure 6B). The absence of this motif would explain why wsMrap1 failed to facilitate the activation of paddlefish Mc2r (Figure 4B). The N-terminal domains of esMrap1 and wsMrap1 do have an N-linked glycosylation consensus sequence (i.e., N-X-S) near the N-terminal, which is also present in the osteichthyan Mrap1 sequences. Likewise, the Y/F-E-Y-Y motif is also present in the N-terminal domains of the cartilaginous fish Mraps and all later-derived gnathostome Mraps. However, in our examination, the sequence identity/similarity between the N-terminal domain for the Mrap1 sequences of cartilaginous fishes and osteichthyans was only 34% (Figure 6). By contrast, the transmembrane domains of the osteichthyan and cartilaginous fish Mrap1 sequences have much greater sequence identity/similarity (Figure 6), which is consistent with the conserved role that Mrap1 has in Mc2r membrane-trafficking in both of these lineages.

The origins of Mraps in the vertebrate lineage remains unresolved. Presumably, an *mrap1*-like gene was present in the ancestral gnathostomes (Figure 5) and would have diverged considerably during the radiation of the ancestral cartilaginous and osteichthyan lineages, in parallel with the remarkable divergence of the *mc2r* gene during this particularly plastic phase of vertebrate evolution. It should be noted that the current designation of the cartilaginous Mrap sequence as an “Mrap1” is based on the sequence similarities obtain from BLAST analysis (NCBI). It is curious that these cartilaginous fish Mrap1 sequences align with the N-terminal and transmembrane domains of the primary sequence of the Mrap sequence available for sea lamprey (*Petromyzon marinus*), a jawless vertebrate, with a higher primary sequence identity/similarity (42%) than to mouse Mrap1 (32%) (Figure 6). The current designation of the Mrap sequence for *P. marinus* is an “Mrap2” because of its apparent primary sequence identity with osteichthyan Mrap2 orthologs and its lack of a δ-D-Y-δ motif in the N-terminal domain. However, the *P. marinus* lineage is much older than any of the cartilaginous fish or osteichthyan fish lineages [16], and may be more representative of the organizational plan of the ancestral Mrap (Figure 7).

The timing of the first (1R) and second (2R) whole-genome duplication events in relation to the radiation of the ancestral agnathan lineage is still not fully resolved [33,34,35,36]. However, given the presence of *mrap*-like or *mrap* orthologs in the agnathan and early gnathostome lineages, it is clear that an ancestral *mrap*-like gene (which we refer to here and in Figure 7 as “*mrap-prime*”) was present prior to the 2R event that predated the radiation of the earliest extant gnathostome lineages. Based on the available agnathan and gnathostome Mrap sequences, the primary protein structure of this ancestral Mrap-prime is predicted to have had a N-terminal domain with a reverse topology motif and a transmembrane domain with primary sequence identity/similar to the corresponding domains in extant lamprey Mrap (Figure 6B).

We hypothesize that following the 2R genome duplication event, two paralogous *mrap* genes emerged (referred to as “*mrap-prime a*” and “*mrap-prime b*”), and that these two *mrap* paralogs would have been present in the ancestral gnathostomes (Figure 7). In this scenario, the functional interaction with the ancestral gnathostome Mc2r would have had its origins during this period. We hypothesize that ancestral gnathostome Mc2r ortholog present during this period would have been a generic melanocortin receptor that could have been activated by either ACTH or MSH-related ligands. Following the divergence of the ancestral gnathostomes, the extant cartilaginous fish lineages still retained the Mrap-prime A and Mrap-prime B, which are Mc2r accessory proteins that can facilitate trafficking of the elasmobranch Mc2r [13] but which lack δ-D-Y-δ activation motif, and which we are now designate as “Mrap1” and “Mrap2”, respectively. However, while the Mc2r ortholog of extant elasmobranchs has retained the generic ligand selectivity of the ancestral gnathostome Mc2r, it appears the elephant shark *mc2r* gene had accumulated mutations which allowed this Mc2r to traffic to the plasma membrane without requiring assistance from Mrap [19]. Finally, during the early radiation of the ancestral osteichthyan fishes, mutations accumulated in both the *mc2r* gene and one of *mrap-prime* genes, resulting in one of the *mrap* paralogs being transformed into what we now consider as a bona fide *mrap1* gene (i.e., presence of the δ-D-Y-δ activation motif) and interacts with Mc2r to render it exclusively selective for ACTH. The other *mrap* paralog of the ancestral osteichthyans would eventually become what we now consider Mrap2 and develop an interaction with Mc4r (for review see [37]). This hypothetical scheme underscores the dynamic changes that occurred to both the *mc2r* gene and the *mrap* genes during the early radiation of the gnathostomes.

## 5. Conclusions

During the early radiation of the gnathostomes the *mc2r* gene and the *mrap* genes (the forerunners of the *mrap1* and *mrap2* genes) underwent a number of modifications with unique outcomes for extant cartilaginous fishes and extant osteichthyan vertebrates. For the cartilaginous fishes, the Mc2r ortholog present on interrenal cells in the hypothalamus/pituitary/interrenal axis can be activated by either ACTH or MSH-related ligands. It appears that for elasmobranch Mc2r orthologs, interaction with an Mrap is required to facilitate the trafficking of the receptor from the endoplasmic reticulum to the plasma membrane. A distinctive feature of the elasmobranch “Mrap1” ortholog is that it lacks the δ-D-Y-δ activation motif. In addition, at least for one holocephalon cartilaginous fish (elephant shark), interaction with an Mrap is not required for trafficking. These outcomes are in sharp contrast to the changes that occurred in the *mc2r* gene of the osteichthyan vertebrates which rendered this Mc2r ortholog exclusively selective for ACTH. In addition, mutations in the osteichthyan *mrap1* gene that resulted in the acquisition of the δ-D-Y-δ activation motif in the N-terminal domain play a significant role in ligand selectivity and activation of osteichthyan Mc2r orthologs. Hence, the interaction between Mc2r and Mrap1 is a critical regulatory point in the functionality of the hypothalamus/pituitary/adrenal-interrenal axis of osteichthyan vertebrates.

## Figures and Tables

**Figure 1 biomolecules-12-01681-f001:**
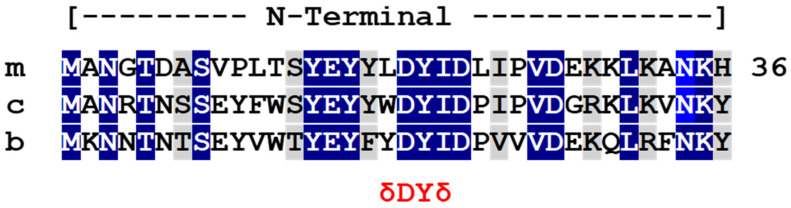
Multiple amino acid sequence alignment of the N-terminal domain of osteichthyan Mrap1s. Mrap1 sequences of mouse (m), chicken (c), and bowfin (b) were aligned, and primary sequence identity (highlighted blue) and primary sequence similarity (highlighted gray) were determined using BLOSUM (https://www.ncbi.nlm.nih.gov/Class/FieldGuide/BLOSUM62.txt) accessed on 6 August 2022. The overall sequence identity for the three orthologs was 44% and the overall sequence similarity was 64%. Each Mrap1 ortholog contained a putative a δ-D-Y-δ (activation) motif [7].

**Figure 2 biomolecules-12-01681-f002:**
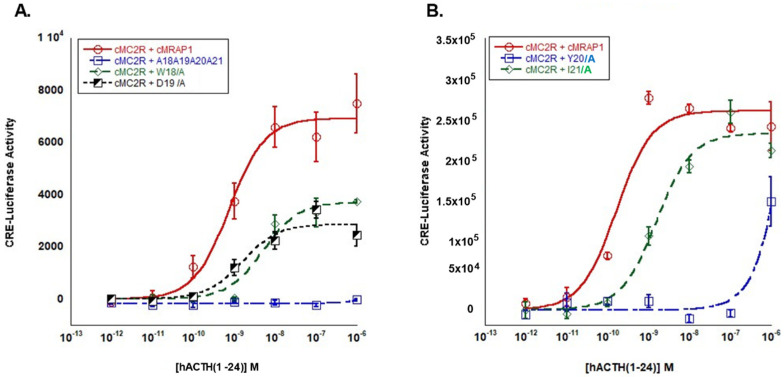
Analysis of W^18^D^19^Y^20^I^21^ motif in chicken MRAP1. (**A**) Activation of chicken (c) MC2R co-transfected with either wildtype *cMRAP1* (positive control) or alanine substitution mutants of *cMRAP1*: *cMRAP1*−*A^18^A^19^A^20^A^21^*, *cMRAP1*−*W^18^/A*, or *cMRAP1*−*D^19^/A*. (**B**) Activation of cMC2R co-transfected with either wildtype *cmrap1* (positive control) or alanine substitution mutants of *cMRAP1*: *cMRAP1*−*Y^20^/A* or *cMRAP1*−*I^21^/A*. Receptor activation was assessed after stimulation with hACTH(1-24). Data presented as mean ± SEM (*n* = 3).

**Figure 3 biomolecules-12-01681-f003:**
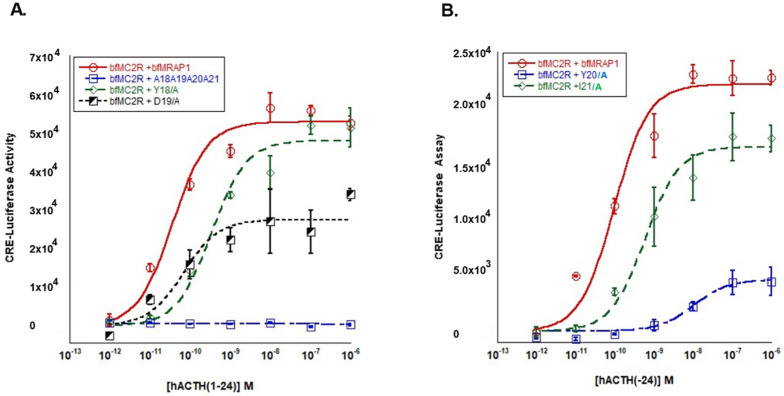
Analysis of Y^18^D^19^Y^20^I^21^ motif in bowfin Mrap1. (**A**) Activation of bowfin (bf) Mc2r co-transfected with either wildtype *bfmrap1* (positive control) or alanine substitution mutants of *bfmrap1*: *bfmrap1-a^18^a^19^a^20^a^21^*, *bfmrap1*−*y^18^/a*, or *bfmrap1*−*d^19^/a*. (**B**) Activation of bfMc2r co-transfected with either wildtype *bfmrap1* (positive control) or alanine substitution mutants of *bfmrap1*: *bfmrap*−*-y^20^/a* or *bfmrap1*−*i^21^/a*. Receptor activation was assessed after stimulation with hACTH(1-24). Data presented as mean ± SEM (*n* = 3).

**Figure 4 biomolecules-12-01681-f004:**
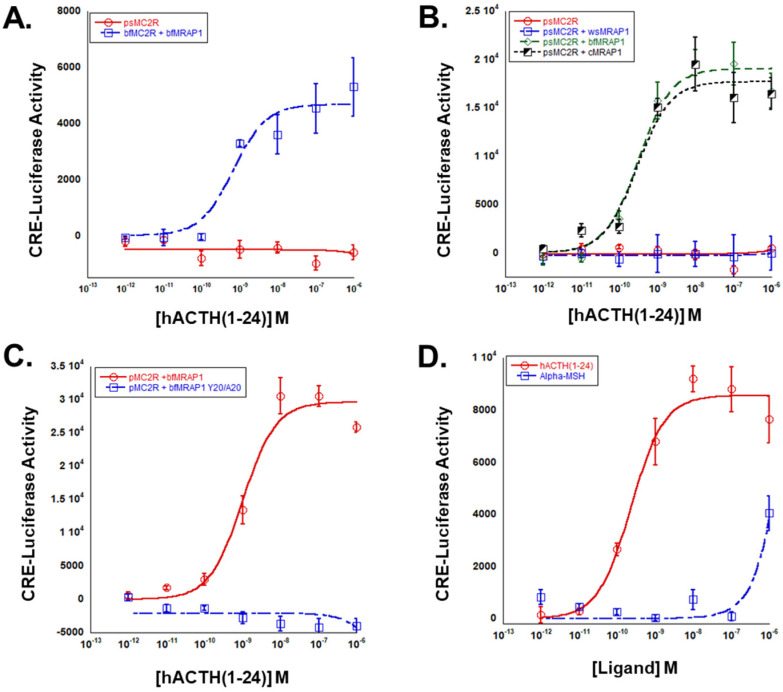
Functional analysis of the paddlefish Mc2r. (**A**) When pMc2r was expressed alone, there was no activation of the receptor by hACTH(1-24). The positive control for this assay was bowfin (bf) Mc2r co-expressed with bfMrap1. Note the robust activation following stimulation with hACTH(1−24). (**B**) Analysis of pMc2r stimulation by hACTH(1-24) when co-expressed with various vertebrate Mrap1s: whale shark (ws) Mrap1, bfMrap1, or chicken (c) MRAP1. (**C**) Analysis of pMc2r stimulation by hACTH(1-24) when co-expressed with either bfMrap1 (positive control) or a bfMrap1-Y^20^/A^20^ mutant. (**D**) Analysis of ligand selectivity of pMc2r, wherein pMc2r was co-expressed with bfMrap1 and stimulated with either hACTH(1-24) or α-MSH. Data presented as mean ± SEM (*n* = 3).

**Figure 5 biomolecules-12-01681-f005:**
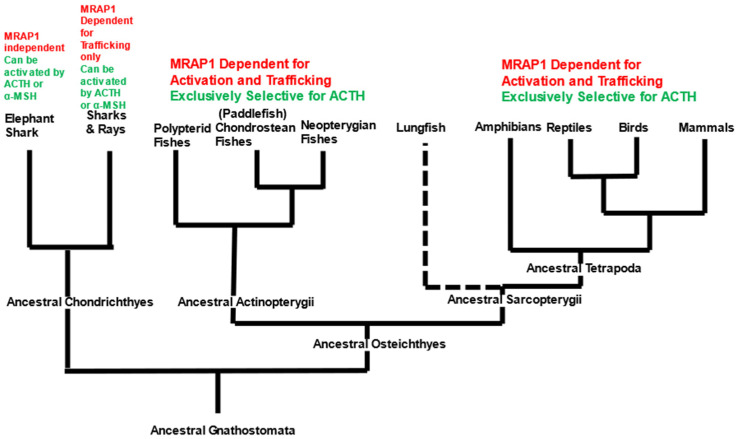
Summary of Mc2r activation requirements in gnathostomes. Definition of terms: Gnathostomata, is bilaterally symmetrical with a vertebral column and true hinged jaws; Osteichthyes, has bony vertebral column; Sarcopterygii, all bony vertebrates with lobed fins or limbs; Actinopterygii, all bony fishes with ray fins; Tetrapoda, all vertebrates with four limbs; Chondrichthyes, gnathostomes with a cartilaginous vertebral column. Dashed line highlights that functional studies have not been performed on the Mc2r ortholog of the basal Sarcopterygian lineages, such as the lungfishes.

**Figure 6 biomolecules-12-01681-f006:**
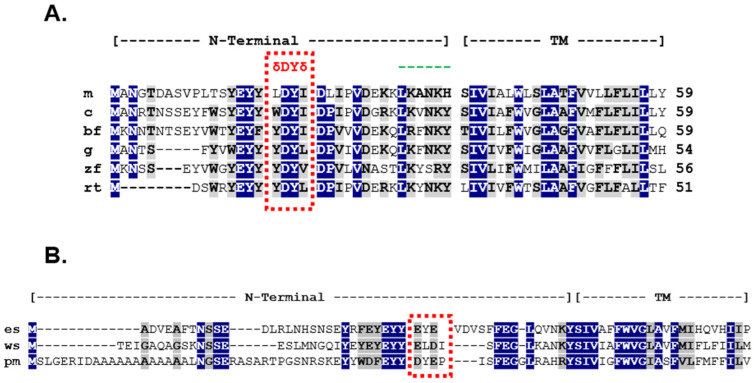
Multiple amino acid sequence alignment of vertebrate Mrap orthologs. (**A**) The amino acid sequence of N-terminal and transmembrane domains of mouse (m) Mrap1 (Accession No. NP_084120) chicken (c) MRAP1 (XR_001470382), bowfin (bf) Mrap1 (LOC: AMCT00016091, [17]), zebrafish (zf) Mrap1 (XP_001342923), and rainbow trout (rt) Mrap [accession number FR837908] were aligned and primary sequence identity (highlighted blue) and primary sequence similarity (highlighted gray) were determined using BLOSUM (https://www.ncbi.nlm.nih.gov/Class/FieldGuide/BLOSUM62.txt) accessed on 6 August 2022. (**B**) The amino acid sequence of the N-terminal and transmembrane domains of elephant shark (es) Mrap1 (XM_007903550.1), whale shark (ws) Mrap1 (XP_020375601), and sea lamprey, *Petromyzon marinus* (pm) Mrap2 (BR000864.1) were aligned and analyzed as described above.

**Figure 7 biomolecules-12-01681-f007:**
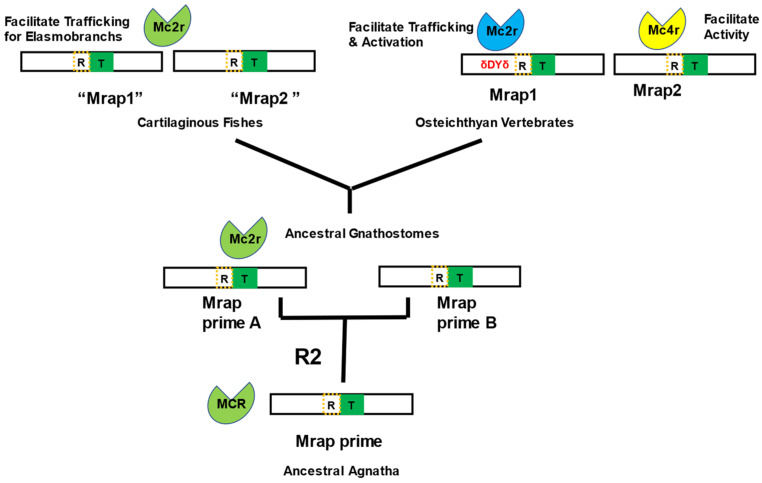
Proposed evolution of Mrap during the radiation of vertebrates. See text for explanation and discussion. Abbreviations: R—reverse topology motif in the N-terminal of Mrap; T—transmembrane domain in Mrap; R2—second chordate genome duplication event.

**Table 1 biomolecules-12-01681-t001:** Analysis of EC_50_ values derived from the experiments presented in Figure 2 and Figure 3.

Panel	Transfection with *cmc2r*	EC_50_ (M)	*p*
2A	*cMRAP1 (positive control)*	7.6 × 10^−10^ ± 2.0 × 10^−10^	
2A	*cMRAP1*−*A^18^A^19^A^20^A^21^*	n.d.	
2A	*cMRAP1*−*W^18^/A*	5.1 × 10^−10^ ± 12.1 × 10^−9^	0.06
2A	*cMRAP1*−*D^19^/A*	1.6 × 10^−9^ ± 8.1 × 10^−10^	0.07
2B	*cMRAP1 (positive control)*	1.6 × 10^−10^ ± 7.7 × 10^−10^	
2B	*cMRAP1*−*Y^21^/A*	n.d.	
2B	*cMRAP1*−*I^22^/A*	1.5 × 10^−9^ ± 5.2 × 10^−10^	0.11
3A	*bfmrap1 (positive control)*	3.6 × 10^−11^ ± 1.1 × 10^−10^	
3A	*bfmrap1*−*a^18^a^19^a^20^a^21^*	n.d.	
3A	*bfmrap1*−*y^18^/a*	3.3 × 10^−10^ ± 1.3 × 10^−10^	0.07
3A	*bfmrap1*−*d^19^/a*	6.8 × 10^−11^ ± 4.1 × 10^−11^	0.23
3B	*bfmrap1 (positive control)*	9.5 × 10^−11^ ± 3.1 × 10^−11^	
3B	*bfmrap1*−*y^20^/a*	1.9 × 10^−8^ ± 4.2 × 10^−9^	0.04
3B	*bfmrap1*−*i^21^/a*	5.8 × 10^−10^ ± 1.6 × 10^−10^	0.02

*p* values reflect comparison to positive control within each experiment as determined by one-way ANOVA (2A, 3A) or Student’s *t*-test (2B, 3B). Data presented as mean ± SEM; n.d. = not determined.

## Data Availability

All data pertaining to this study are stored in the laboratory of Robert M. Dores 613 (robert.dores@du.edu) and are available upon request.

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
