# Peer review of "Structure/Function Studies on the Activation Motif of Two Non-Mammalian Mrap1 Orthologs, and Observations on the Phylogeny of Mrap1, Including a Novel Characterization of an Mrap1 from the Chondrostean Fish, Polyodon spathula"

_biomolecules, 2022, doi:10.3390/biom12111681_

Round 1

Reviewer 1 Report

General comments

The authors presented a very interesting paper on the functionality of MRAP1 in chicken, neopterygian ray-finned fish and paddlefish. Using alanine substitution, they showed the importance of the amino acids of the binding sites in chicken and bowfin for the activity of MC2R. They further showed that paddlefish needs a MRAP1, even if not yet characterized in this species.

Minor comments

L 95: Reference for methods of alanin substitution.

L292: the sentence is not finished

L315-324: is the evolution of MC2R linked to the evolution of the POMC genes, especially with the delta-MSH in cartilogenous fish?

Figure 2: There is a huge difference in the maximal CRE-luciferase activity for the wild-type cMC2R + cMRAP1 between part 2A and 2B, and there is a huge variation in the mean value of the EC50 for 2B positive control: 1.6 *1E-10+- 7.7 *1E-10 for the SEM, a SD of 1.33 *1E-9. Since the mutated values are reported to the control, it may be worse to repeat the experiment to be sure it is correct or is it a typing mistake.

Author Response

General comments

The authors presented a very interesting paper on the functionality of MRAP1 in chicken, neopterygian ray-finned fish and paddlefish. Using alanine substitution, they showed the importance of the amino acids of the binding sites in chicken and bowfin for the activity of MC2Rhey further showed that paddlefish needs a MRAP1, even if not yet characterized in this species.

We thank the reviewer for these comments.

Minor comments

L 95: Reference for methods of alanin substitution.

This was an oversight. The sentence that begins on line 94 now reads, “Several alanine-substituted mutants of bfmrap1 and cMRAP1 were made to target the residues of the putative activation motifs (Fig. S1A-B), and the cDNA constructs for each cmrap1 mutant and bfmrap1 mutant were synthesized by GenScript (Piscataway, NJ).”

L292: the sentence is not finished

This is a typo from an earlier draft the clause has been removed.

L315-324: is the evolution of MC2R linked to the evolution of the POMC genes, especially with the delta-MSH in cartilogenous fish?

It would be reasonable to conclude that the melanocortin receptors have co-evolved with the pomc gene. One of the remarkable features of the pomc gene is that the organizational plan for the prohormone has remained relatively constant for the gnathostomes (i.e., the ACTH/α-MSH sequence in the middle of the precursor and the β-endorphin sequence at the C-terminal of the precursor). In addition, based on studies of cartilaginous fish POMC sequences and bony vertebrate POMC sequences, it would be reasonable to assume that the γ-MSH, ACTH/α-MSH, β-MSH, and β-endorphin sequences were present in the ancestral gnathostome pomc gene (see Vallarino et al., 2012, G.C.E. 177, 353-364). Changes to the precursor include the secondary addition of the δ-MSH sequence in cartilaginous fish POMC orthologs and the secondary loss of  the γ-MSH sequence from teleost POMC orthologs. Although there have been attempts to match a particular  melanocortin receptor with a specific MSH-related ligand, these studies have not been conclusive. The only consistent one-to-one ligand/receptor match is the ACTH/Mc2r relationship for bony vertebrates.  

Figure 2: There is a huge difference in the maximal CRE-luciferase activity for the wild-type cMC2R + cMRAP1 between part 2A and 2B, and there is a huge variation in the mean value of the EC50 for 2B positive control: 1.6 *1E-10+- 7.7 *1E-10 for the SEM, a SD of 1.33 *1E-9. Since the mutated values are reported to the control, it may be worse to repeat the experiment to be sure it is correct or is it a typing mistake.

Figure 2B is not a typing mistake. All the activation assays presented in this manuscript involve transient transfections of CHO cells. Variability between assays on different days will be a function of the “age” of the CHO cells (i.e., the number of trypsin passages). After doing these assays for the past 12 years we have found the response of the CHO will show a decline after a number of passages. Hence, we stop using CHO cells after the 17th passage. In addition, the maximum response will shift if more cells are used on a given day relative to a previous day’s experiment.

The critical issue, however, for assays on each day is the positive control (i.e, MC2R + wild type MRAP1.) All transfections involving an experimental manipulation are compared to the positive control.  For a given MC2R such as chicken (c) MC2R, the EC50 values (measure of sensitivity) should not vary statistically between assays. For Figure 2A the EC50  value for the positive control was 7.6 x 10-10M + 2.0 x 10-10 and for Figure 2B the EC50 value for the positive control was 1.6 x 10-10M + 7.7 x 10-10 (n =3). Student t-Test analysis indicates a p value of 0.23. These EC50 values are not statistically different. However, what is apparent in Figure 2B is the dramatic shift and in EC50 value  and Vmax value for the cMRAP1 Y20/A dose response curve. Co-transfecting cMC2R with this mutant form of cMRAP1 has a significant effect on activation, and that was the point we wanted to make in Figure 2B.

Figure 2B  is a repeat of this assay. In the previous attempt, the student saw the same relationships (i.e., cMC2R/cMRAP1 I21/A behaved in a very similar manner to the positive control and cMC2R/cMRAP1Y20/A showed a dramatic shift in EC50 and Vmax values. However, I did not feel that the precision of the assay was journal quality. The student repeated the experiment with a fresh batch of CHO cells from our stock of frozen cells, but clearly used more CHO cells per transfection as indicated by the Vmax values for the dose response curves in Figure 2B as compared to the dose response curves for Figure 2A as the reviewer observed.

Would repeating the experiment a 3rd time result in a “worse” result as the reviewer queried? I am confident that we would see the same trend; cMRAP1 Y20/A interferes with activation. We could normalize the results relative to the positive control if the editor feels this manipulation is needed. It will not change the shape of the dose response curve nor the conclusions. If the editor feels that we should make this adjustment, then we will make the same adjustment to Figure 2A, Figure 3, and Figure 4 in the next revision of this manuscript

Reviewer 2 Report

The study is interesting, providing insights into the evolution of Mrap with respect to its apparent interactions with the MC2R.

Minor points:

1. For MRAP mutants, they (W18A/A18, D19/A19, Y20/A20, or I21/A20) should be changed to W18A, D19A, Y20A, or I21A respectively.

2, Did you detect these MRAP1 mutants on MC2R ligand binding and cell surface expression?

3. Line 292-293 incomplete sentence

4. Unclear figure 4

Author Response

The study is interesting, providing insights into the evolution of Mrap with respect to its apparent interactions with the MC2R.

 We thank the reviewer for these observations.

Minor points:

  1. For MRAP mutants, they (W18A/A18, D19/A19, Y20/A20, or I21/A20) should be changed to W18A, D19A, Y20A, or I21A respectively.

All of the mutant Mrap1 designations have been changed in the manuscript, following the instructions of the reviewer.

  1. Did you detect these MRAP1 mutants on MC2R ligand binding and cell surface expression?

We did not do binding assays in this study. That analysis is planned for a future study.

Since the mutations were made in the N-terminal domain of the respective Mrap1 ortholog, we did not look at surface expression. We assumed that since the transmembrane domain of each mutant was unaltered that that trafficking would not be affected. However, the N-terminal of cMRAP1 and bfMrap1 is most likely interacting with Extracellular Loop 2 of the respective receptor. Investigating that possibility would be part of future studies on these receptors. 

  1. Line 292-293 incomplete sentence.

This is a typo from an earlier draft the clause has been removed.

  1. Unclear figure 4

We understand the problem. The description for Legend 4A) was incoherent. A) now reads, “When pMc2r was expressed alone, there was no activation of the receptor by hACTH(1-24). The positive control for this assay was bowfin (bf) Mc2r co-expressed with bfMrap1. Note the robust activation following stimulation with hACTH(1-24).”